# The Roles of Microbial Cell-Cell Chemical Communication Systems in the Modulation of Antimicrobial Resistance

**DOI:** 10.3390/antibiotics9110779

**Published:** 2020-11-06

**Authors:** Ying Huang, Yufan Chen, Lian-hui Zhang

**Affiliations:** 1Guangdong Laboratory for Lingnan Modern Agriculture, Guangzhou 510642, China; huangying@stu.scau.edu.cn (Y.H.); chenyufan@stu.scau.edu.cn (Y.C.); 2Guangdong Province Key Laboratory of Microbial Signals and Disease Control, Integrative Microbiology Research Center, South China Agricultural University, Guangzhou 510642, China

**Keywords:** antibiotics, antimicrobial resistance, quorum sensing, host-pathogen communication, quorum quenching

## Abstract

Rapid emergence of antimicrobial resistance (AMR) has become a critical challenge worldwide. It is of great importance to understand how AMR is modulated genetically in order to explore new antimicrobial strategies. Recent studies have unveiled that microbial communication systems, which are known to play key roles in regulation of bacterial virulence, are also associated with the formation and regulation of AMR. These microbial cell-to-cell chemical communication systems, including quorum sensing (QS) and pathogen–host communication mechanisms, rely on detection and response of various chemical signal molecules, which are generated either by the microbe itself or host cells, to activate the expression of virulence and AMR genes. This article summarizes the generic signaling mechanisms of representative QS and pathogen–host communications systems, reviews the current knowledge regarding the roles of these chemical communication systems in regulation of AMR, and describes the strategies developed over the years for blocking bacterial chemical communication systems in disease control. The research progress in this field suggests that the bacterial cell-cell communication systems are a promising target not only for disease control but also for curbing the problem of microbial drug resistance.

## 1. Introduction

Pathogenic microorganisms are one of the major threats to the life quality of living organisms, including humans, animals and plants. In the pre-antibiotic era, microbial infections accounted for high morbidity and high mortality of human beings worldwide. Virulent organisms were capable of spreading infections at a very rapid rate and used to cause widespread outbreak, epidemics, or pandemics. Little progress had been made in finding effective ways to prevent and treat microbial infections until the late 19th century, when Robert Koch and Louis Pasteur discovered that microbes are the cause of various infections [1,2]. In 1909, Paul Ehrlich, a German physician and scientist, discovered the first effective medicinal treatment for syphilis, called arsphenamine (salvarsan) [3]. In 1929, Alexander Fleming isolated the first broadly effective antibiotic, benzylpenicillin (penicillin G), from the filtrate of a fungal culture, marking the beginning of a golden age in antibiotic research and infection control [4]. In 1947, the word “antibiotics” was coined by the Ukrainian-American inventor and microbiologist Selman Waksman, who discovered streptomycin and several other antibiotics in his lifetime [5]. There is no doubt that the discovery of antibiotics is a great milestone in the history of human anti-infection, which has significantly increased the life span of humanity internationally [5,6,7].

However, large-scale and improper use of antibiotics inevitably lead to the growing problem of antimicrobial resistance (AMR). The antibiotic-resistant or drug-resistant bacteria can be classified as multidrug-resistant (MDR), extensively drug-resistant (XDR), and even pandrug-resistant (PDR) strains [8]. AMR has become the biggest obstacle for antibiotic treatment of pathogenic infections, which is responsible for about 700,000 fatalities each year and about 1% of world GDP losses currently. By 2050, it was estimated that AMR may cause about 10 million deaths with up to USD 100 trillion in global economic losses [9]. It is believed that development of antibiotic resistance is mainly due to the following factors: firstly, lack of stewardship of many antimicrobial agents in health care, agriculture and veterinary sectors [10]; secondly, certain AMR mechanisms are ancient and existed prior to the use of antibiotics [11]; furthermore, genetic determinants of AMR are often transferable, which aids the spread of AMR by horizontal, intra-species and inter-species migration [12]. In addition to the above reasons, many antimicrobials target basic pathways such as DNA and protein synthesis that are common to microorganisms, exerting strong selection pressures in favor of resistance mutations [13].

Interestingly, AMR has been found to be regulated by multiple mechanisms, including two-component systems (TCS), transcription factors, quorum sensing (QS) systems, bacterial second messenger molecules, etc. For example, bacterial efflux pumps, which could efflux multiple antibiotics [14], are regulated by various transcriptional repressors, global regulators and TCSs [15]. In *Escherichia coli*, the TCS ArcAB activates the expression of efflux pump genes *mdtEF* under anaerobic conditions through binding to the promoter region of the *gadE-mdtEF* operon [16]. In *Pseudomonas aeruginosa*, the ubiquitous bacterial second messenger cyclic di-GMP positively regulates the production of extracellular matrix components such as adhesin CdrA, exopolysaccharides alginate, Pel and Psl [17]. Along with c-di-GMP, the small regulatory RNAs (sRNA) were also reported to regulate biofilm formation and dispersal in several bacterial species [18]. Recent studies have shown that a microbial cell–cell communication mechanism, known as QS, is associated with modulation of bacterial virulence as well as a variety of AMR mechanisms. In *P. aeruginosa*, a complex hierarchy QS network not only controls the formation of biofilm, which is one of the important AMR mechanisms as summarized in the following section, but also regulates the expression of the drug efflux pump system gene *mexAB-oprM* [19,20]. In contrast, the diffusible signaling factor (DSF)-type QS system, originally identified in plant pathogen *Xanthomonas campestris*, plays a critical role in positive regulation of biofilm dispersal through modulation of the production of exopolysaccharide encoded by the *xagABC* gene cluster [21]. Importantly, the QS system could act as an upstream signaling system to control bacterial AMR mechanisms by regulating intracellular second messenger concentrations and sRNA transcription, as well as expression of transcription factors [22,23].

Rapid increase in AMR problems highlights the need of searching for new strategies or alternative approaches to prevent formation and spreading AMR, or sensitize pathogenic microorganisms to conventional antibiotics. In this regard, understanding the potential molecular mechanisms associated with modulation of AMR may hold the key to unlock the AMR problem. Since 1990s, it has become known that numerous microbial activities, including plasmid conjugal transfer, virulence factor production, biofilm formation, and expression of efflux pumps, are regulated by microbial cell-cell communication mechanisms, including QS and pathogen–host chemical signaling systems [24,25,26,27]. Among them, several microbial functions are associated with AMR, such as plasmid horizontal transfer, biofilm formation and generation of efflux pumps. In this review, we will summarize the known AMR mechanisms, discuss the fundamentals of QS and pathogen–host signaling systems, and review the roles of microbial chemical communication systems in regulation of AMR mechanisms to counteract antibiotics. Significantly, evidence is emerging that blocking these microbial cell-cell communication systems could provide effective control against microbial infections, highlighting a promising potential to also curb the emerging AMR problems.

## 2. Overview of Antimicrobial and AMR Mechanisms

Conventional antibiotics act by stopping microbial growth or killing microorganisms through either inhibiting microbial primary metabolic processes or compromising pathogen cell–membrane integrity. Different types of antibiotics that target various cellular processes have been discovered and used in clinical practice. Numerous studies unveiled that microorganisms could resist to antibiotics through various mechanisms, including cell wall rebuilding, efflux pump, target mutation, enzymes for degradation or modification of antibiotics, or alteration of membrane permeability. In addition, bacterial cells can form biofilms in which numerous attached cells are encased within extracellular polymeric matrix composed of mainly polysaccharides, lipids, proteins and extracellular DNA, thus enabling microbial cells to withstand the entry and action of antibiotics [28,29,30,31,32,33,34,35,36,37,38].

Specifically, the widespread use of antibiotics has increased selection pressure, and mobile genetic elements such as plasmids, integrons, and transposons tend to cluster together to form mobile drug-resistant units. These units have an extraordinary ability to spread among bacteria, allowing for hundreds or thousands of times higher increment in drug-resistance rate and a genetic identity in the subsequent proliferated bacterial populations. It has been found that bacterial colonies in soil are able to acquire gene fragments from other colonies and even transgenic plants through broad-host-range plasmids, highlighting the ability of resistance genes to spread rapidly between colonies and even across species [39,40].

In addition, bacterial biofilm constitutes an effective and complexed AMR mechanism, which has various modes of action [41,42]: (1) the barrier—the glycocalyx or exopolysaccharide component of biofilm can significantly reduce the permeability and sensitivity of antibiotics [43,44]; (2) detoxification—biofilm-producing bacteria can produce enzymes to disrupt or alter the structure of antibiotics to inactivate the antibiotic and drug efflux pumps and can transport antibiotics to outside of the cell, reducing the intracellular concentration of antibiotics to non-sensitive levels [45,46]; (3) heterogeneity—the special microenvironment within the biofilm can alter the metabolic activity and growth rate of bacterial cells, and it accelerates the production of tolerant cells (persisters), which promote AMR development [47,48].

## 3. Fundamentals of Microbial Chemical Communication Systems

The microbial cell–cell chemical communication systems include QS and host–pathogen communication systems. The research progress over the last two decades showed that microbial chemical communication systems play important roles in regulation of a variety of biological processes, such as antibiotic synthesis, plasmid conjugal transfer, virulence gene expression, drug efflux pumps, bioluminescence, and biofilm formation [22,23,49,50]. Identification of these communication systems allow us to understand how microbial virulence and AMR are modulated, and it provides valuable insights on developing new strategies to curb AMR and control infectious diseases.

The origin of QS research can be traced back to at least the mid-20th century, when researchers discovered that uptake of exogenous DNA molecules by *Streptococcus pneumoniae* and production of biofluorescence by the marine bacterium *Aliivibrio fischeri* (also called *Vibrio fischeri*) were related to the bacterial population density, which was called the cell density-dependent phenomenon [51,52]. In 1981, it was identified that *A. fischeri* produces a small chemical signal called autoinducer (AI), which is an acyl homoserine lactone (AHL) containing a 6-carbon fatty acid side chain, to regulate cell density-dependent production of biofluorescence [53]. In 1991, Zhang and Kerr [54] reported that *Agrobacterium tumefaciens* produces a diffusible conjugation factor to induce Ti plasmid conjugal transfer in a population-dependent manner. Two years later, the conjugation factor was identified as an AHL family signal containing eight carbon side chains, and, thus, they proposed that AHL-like derivatives may be a widely conserved signal molecules in microbial community that regulate different biological functions [55]. In 1994, Fuqua et al. [56], coined the term of “quorum sensing” to describe an environmental sensing system that monitors the population density to coordinate the social behaviors of microorganisms. QS regulatory systems are characterized by the fact that microorganisms produce and release to the surrounding environment a diffusible autoinducer or QS signal, which accumulates along with bacterial growth and induces target gene transcriptional expression when reaching a threshold concentration. Typically, a QS system contains a signal synthase, which produces QS signals, and a signal receptor that detect and response to QS signal in a population-dependent manner [23,56,57]. The signal receptor could be a transcription factor such as the LuxR of *A. fischeri* [56], or a two-component system such as the RpfC-RpfG in *X. campestris* [22]. Some microorganisms may contain only one QS system, but others may evolve multiple QS systems. For example, *P. aeruginosa* possesses a hierarchy QS network that uses multiple QS signaling molecules to coordinate the expression of virulence genes and the formation of biofilms [58]. A range of QS signals has been identified and characterized, including AHLs [23], DSFs [22], autoinducer-2 (AI-2) [59], and *Pseudomonas* quinolone signal (PQS) [60].

In addition, microbial pathogens can also exploit the chemical molecules produced by host organisms as cross-kingdom signals to regulate virulence gene expression and traits associated with AMR. The known host signals include plant phenolic compounds that induce T-DNA transfer in *A. tumefaciens* [61], spermidine involved in the regulation of the type III secretion system (T3SS) in *P. aeruginosa* [62], and indole and indole-3-acetic acid associated with the modulation of sexual mating in *Sporisorium scitamineum* [63]. Most of these QS and host signals are involved in the regulation of bacterial virulence and drug resistance [64,65].

### 3.1. AHL-Type QS System

The AHL-mediated QS system is one of the most studied cell-to-cell communication systems, with over 200 bacterial species known to produce AHL family signals. Different bacterial species produce similar AHL signaling molecules with a conserved *N*-acyl homoserine lactone, but the fatty acid side chain lengths and substituents are in general variable, which may account for their signaling specificity in different microorganisms [66]. A wide range of microbial biological functions are known to be regulated by the AHL QS system, including bioluminescence, plasmid DNA transfer, production of pathogenic factors, biofilm formation, and antibiotic production [23]. The core of the AHL QS system consists of the AHL synthase (LuxI homologue) and the AHL signal receptor (LuxR homologue). In the bioluminescent bacterium *A. fischeri*, where the AHL QS system was first identified, at low cell density, the LuxI protein produces a basal level of few diffusible AHL molecules, and little bioluminescence could be detected. In contrast, with an increase in bacterial population density, the AHL signal molecules that accumulated around the bacterial cells enter the cell and bind and activate the LuxR protein, which then initiates LuxI overexpression and activates transcription of luminescence genes [67]. The generic mechanism of AHL-mediated QS is illustrated in Figure 1.

### 3.2. DSF-Type QS System

DSF represents another class of QS systems that are widespread in Gram-negative bacteria. In 1997, the Daniels Laboratory in the UK discovered that plant pathogen *Xanthomonas campestris pv.campesitris* (*Xcc*) could produce a diffusible signaling factor (DSF) and regulate production of a number of virulence factors, including proteases, cellulases and extracellular polysaccharides [68,69]. A subsequent study showed that three genes are associated with the DSF QS system, in which the *rpfF* gene may encode a DSF synthase, and the products of *rpfC* and *rpfG* genes could be involved in DSF signaling [69]. In 2004, Wang et al. isolated and identified DSF as cis-11-methyl-2-dodecenoic acid [70]. To date, more than 10 structurally similar DSF family signals have been isolated and identified from Gram-negative bacteria, and over 100 bacterial species were predicted to use DSF to regulate various biological functions [22,71]. The biological functions and signaling mechanisms of the DSF systems have been well studied in *Xcc* and *Burkholderia cenocepacia* [72,73]. In *Xcc*, response to DSF signals leads to activation of the protein kinase RpfC by self-phosphorylation, which activates the phosphodiesterase activity of its cognate response regulator RpfG through phosphorelay; the activated RpfG degrades c-di-GMP molecules and activates the global regulator Clp; Clp directly regulates the transcription of a number of pathogenic genes and indirectly regulates the expression of other pathogenic genes through the downstream transcription factors FhrR and Zur [72]. In *B. cenocepacia*, another DSF family signal, cis-2-dodecenoic acid (BDSF), binds to its major receptor RpfR to reduce the intracellular c-di-GMP to a sufficient level, which stimulate the RpfR-GtrR protein complex to bind to the promoter region of target DNA, thus modulating the production of virulence factors and biofilm mass [74,75,76].

### 3.3. Polyamine-Mediated Host–Pathogen Communication Systems

Spermidine (Spd), spermine (Spm), and putrescine (Put) represent a widespread group of cationic aliphatic compounds with multiple biological activities in living cells, collectively referred to as polyamines. In recent years, increasing evidence indicates that polyamines are capable of performing regulatory functions as signals in intraspecies cell–cell communication or are involved in host–pathogen trans-kingdom cell-cell communication [62,77]. The precursors for synthesis of Put, Spd and Spm in bacterial cells are l-arginine, l-ornithine and l-methionine, respectively [78]. l-arginine decarboxylase decarboxylates arginine to produce adamantane, which is then converted to Put and urea by agmatine ureohydrolase. l-ornithine is decarboxylated by ornithine decarboxylase (ODC) to produce Put, then Put undergoes a transfer reaction with a propylamine group by the action of arginine synthetase to produce Spd, and then undergoes another transfer reaction with a propylamine group by the action of Spm synthetase to produce Spm [79]. l-methionine is catalyzed by l-methionine adenosyltransferase (MAT) to produce S-adenosylmethionine (SAM), which is then decarboxylated by S-adenosylmethionine decarboxylase (SAMDC) to produce decarboxylated S-adenosylmethionine (dcSAM), which provides the propylamine group for the synthesis of Spd and Spm [79]. In bacteria, polyamines are mainly involved in the regulation of important physiological processes such as transcription and translation of bacterial virulence factors, biofilm formation, antibiotic resistance, acidic and oxidative stresses [78,80].

Evidence indicates that polyamines are not only bacterial QS signals, but also play a key role in pathogen–host communications. Polyamine signals accumulated in the environment, either produced by bacterial cells or by host cells, are recognized and fluxed into bacterial cells through membrane associated transporters [26,62,77]. For example, Spd and Spm were identified as host signals for induction of T3SS in *P. aeruginosa*. These two signals are transported into the bacterial cells through a spermidine-specific ABC transporter SpuDEFGH [62]. Crystal structure analysis showed that Spd signals are specifically recognized by the substrate-binding protein SpuE, while another substrate-binding protein SpuD interacts with Put [81]. Deletion of this transporter significantly reduced the transcriptional expression of the T3SS genes, and markedly decreased the T3SS-mediated cell cytotoxicity and virulence [62]. In plant pathogen *Dickeya zeae*, it is Put that plays dual roles as a QS signal and cross-kingdom communication signal in regulation of bacterial motility and biofilm formation [77]. Deletion of *speA* in *D. zeae* that encodes ODC for Put biosynthesis markedly reduced bacterial motility and biofilm formation, which can be rescued by exogenous addition of rice extracts containing Put [77]. Two putrescine-specific transporters in *D. zeae*, i.e., PotF and PlaP, play a key role in the bacterial influx of Put signals produced by themselves or by the host plant. The *D. zeae* null mutant lacking PotF and PlaP was much attenuated in systemic infection and virulence against rice seeds [77].

## 4. The Role of the AHL QS System in Modulation of Antibiotic Resistance

*P. aeruginosa* is a major causative agent of cystic fibrosis (CF), and its AMR mechanisms and related regulatory networks have been well-documented. The pathogen was well-known for its multiple AMR mechanisms, which can be divided into three categories: intrinsic, acquired and adaptive resistance [82]. The intrinsic resistance includes low permeability of extracellular membranes, active efflux pumps and antibiotic inactivating enzymes [83], whereas acquired resistance can be obtained through horizontal gene transfer or mutation of resistance genes, and adaptive resistance involves the formation of biofilms, which act as a diffusion barrier to limit the entry of antibiotics into bacterial cells [84,85]. In 1998, Davies et al. [86], first demonstrated that the AHL QS system is involved in regulation of biofilm formation in *P. aeruginosa*. A subsequent study found that mutant of *lasI*, which encodes an AHL signal synthase, formed flat and undifferentiated biofilms and became more sensitive to the antimicrobial agent sodium dodecyl sulfate (SDS) than the wild type *P. aeruginosa* [86].

The QS network of the *P. aeruginosa* has been found to consist of four signaling systems, namely Las, Rhl, PQS and integrated quorum sensing system (IQS). Among them, Las and Rhl systems produce and response to AHL signaling molecules 3O-C12HSL and C4-HSL, respectively, which induce biofilm formation and expression of various virulence genes by binding and activating their corresponding cognate transcription factors LasR and RhlR, respectively [20]. In addition, exogenous addition of the quorum-sensing molecule C4-HSL could up-regulate the expression of *mexAB-oprM*, which encodes a drug efflux pump system and enhances the resistance of bacteria to various antibiotics, whereas 3O-C12HSL showed no significant effect on the regulation of *mexAB-oprM* gene expression [19,87]. Furthermore, the quorum-sensing signals PQS and 2-heptyl-4-quinolone (HHQ), which are generated by the *pqsABCDEH* operon, regulate the expression of downstream virulence factors and biofilm production by binding to the transcriptional regulator PqsR (MvfR) [60,88].

## 5. The Role of DSF QS System in Modulation of Antibiotic Resistance

In contrast to the AHL QS system, which regulates biofilm formation in a positive manner, DSF QS systems appear to play a negative role in modulation of bacterial biofilm formation. This is evident from the studies on *Xcc*, *B. cenocepacia* and *P. aeruginosa* [22]. In *Xcc*, when bacterial cells grow to a high population density, the accumulated DSF signals interact with the N-terminal 22 amino acid-length sensor region of the histidine kinase (HK) RpfC [89], which activates the cognate response regulator RpfG and releases the global regulator Clp through degradation of c-di-GMP [90]. Clp plays dual roles in regulation of biofilm formation, i.e., inhibiting the expression of *xagABC* encoding a glycosyl transferase involved in biofilm formation, and activating the expression of *manA* encoding endo-β-1,4-mannanase associated with biofilm degradation, thus negatively regulating the bacterial biofilm formation [89,91,92]. Interestingly, genome microarray analysis showed that DSF positively regulates over 10 genes encoding drug resistance and oxidative stress resistance [90], but the impact of DSF signaling on AMR has not yet been experimentally evaluated.

*B. cenocepacia* relies on a DSF family signal, i.e., BDSF, which differs from DSF by lacking a methyl substitution on C11 position [70], to regulate biofilm formation and bacterial virulence [22,76]. The BDSF receptor RpfR undertakes dual functions (signal perception and transmission) of RpfC and RpfG. At low cell density, the second messenger c-di-GMP binds to and thus inactivates the RpfR-GtrR protein complex, whereas at high cell density, RpfR detects the BDSF signal through its PAS domain and causes a conformational change in the RpfR protein, which activates the phosphodiesterase activity of the EAL domain to degrade the intracellular c-di-GMP signal and release the RpfR–GtrR complex to bind to the target promoter DNA, resulting in enhanced production of virulence factors and decreased biofilm formation [73,75]. Nevertheless, the detailed molecular mechanism of BDSF QS signals in regulating bacterial biofilm formation in *B. cenocepacia* remains to be elucidated by further studies.

## 6. The Role of the Polyamine Chemical Communication System in Modulation of Antibiotic Resistance

Microorganisms are able to synthesize polyamines on their own or uptake polyamines from the environment and host via polyamine uptake systems. Evidence is accumulating that suggests that in addition to the role in regulation of bacterial virulence, polyamines also play important roles in the modulation of various stress response mechanisms such as oxidative protection, acid tolerance and AMR (Figure 2) [93]. Polyamine signals seem to influence bacterial antibiotic resistance or susceptibility depending on the chemical properties of antibiotics. In *P. aeruginosa*, exogenously added Spd and Spm were found capable of increasing bacterial resistance to polymyxin B and other cationic peptide antibiotics [94], but also making the pathogen more susceptible to another set of antibiotics, including β-lactams, chloramphenicol, nalidixic acid and trimethoprim [95]. Similarly, results of checkerboard assays with *E. coli* and methicillin-resistant *Staphylococcus aureus* (MRSA) showed that the combination of Spm with β-lactam and chloramphenicol antibiotics had a strong synergistic effect [96].

Genetic analysis showed that exogenous addition of polyamine to *P. aeruginosa* could significantly induce the expression of resistance-associated operon *oprH-phoPQ* and *pmrHFIJKLM*, which encode an outer membrane porin, a two-component regulatory system and the genes for lipopolysaccharide (LPS) modification operon, respectively [94]. This finding may explain why polyamine could increase bacterial resistance to cationic peptide antibiotics, which are known to act by disrupting the structural organization of LPS, thus increasing permeability of the cationic peptide antibiotics and killing bacterial cells. In *E. coli*, polyamines were identified to control ion flow through the porin proteins OmpF and OmpC, and to block the passage of β-lactam antibiotics through these porins, thereby increasing drug resistance [97]. In *B. cenocepacia*, polymyxin B activates the gene encoding for ODC, the first enzyme in the polyamine biosynthetic pathway, resulting in increased levels of intracellular polyamines, which protect bacterial cells against oxidative stress induced by polymyxin B and other antibiotics through enhancing the expression level of the oxidative response regulator OxyR [98]. Similarly, exogenous addition of Put protected against oxidative stress induced by polymyxin B, whereas reduced Put synthesis resulted in increased reactive oxygen species (ROS) generation and a parallel increased sensitivity to polymyxin B [98].

Polyamines may also modulate AMR through regulation of biofilm formation, providing the role of a biofilm in protecting antibiotic penetration. In *Yersinia pestis*, the causative agent of plague, exogenous addition of Put was able to restore the biofilm formation of a polyamine-deficient strain [99]. In the phytopathogen *D. zeae*, deletion of the *speA* gene encoding arginine decarboxylase for Put signal biosynthesis greatly impairs the bacterial cell motility and biofilm formation [77]. These findings plus the fact that polyamine could increase the expression level of LPS modification enzymes in *P. aeruginiosa* [94], suggest that biofilm-mediated AMR might be a generic conserved mechanism regulated by polyamine signals.

## 7. Strategies for Blocking Microbial Chemical Communication Systems

Given the important roles of microbial QS systems in the regulation of bacterial virulence and AMR, it should be feasible to control bacterial infections through developing new strategies to block QS. Shortly after understanding the roles of QS in regulation of bacterial virulence, a novel strategy known as “quorum quenching” has been demonstrated for effectively blocking of bacterial QS or cell–cell communication [100]. In this strategy, a novel AHL-lactonase was identified and used to inactivate AHL QS signal, thus abolishing the AHL-mediated virulence gene expression and bacterial pathogenicity [100,101]. To date, various quorum quenching (QQ) strategies have been developed [66,102,103,104], which can be divided into the following categories: (1) QQ enzymes or antibodies for degradation or modification of QS signals, (2) inhibitors targeting QS signal synthase, and (3) QS signal analogs for competition of signal receptor (Table 1).

Degradation or modification of QS signals means destructing or changing the chemical structure of QS signal molecules by certain enzymes, thus blocking the QS signaling communication. Most of the QQ enzymes that have been reported to date target AHL family QS signals, which have been well-studied since the 1990s, and a range of novel QQ enzymes has been unveiled since the year 2000 [66,102,103]. As for the relatively newly discovered DSF family QS signals, the specific QQ enzymes have only been reported in recent years [107,110]. Depending on the catalytic mechanisms, AHL signals can be completely degraded or inactivated by enzymes, including AHL-lactonases (AiiA), AHL-acylases (AiiD), and AHL oxidoreductases [66,101,102,111]. In 2000, the first QQ enzyme specific for AHL family signals named AiiA was identified from a *Bacillus* species, which hydrolyzes the lactone ring of AHL molecules to produce an acyl-homoserine [101]. A subsequent study demonstrated that expression of AiiA could effectively prevent and control crop soft rot disease caused by *Erwinia carotovora* using transgenic plants [100]. This finding led to identification of other QQ enzymes with different catalytic mechanisms, including AHL-acylase (AiiD), which was found in *Ralstonia* sp., *P. aeruginosa*, and *Variovorax paradoxus* [105,112,113], and AHL-oxidoreductase from *Rhodococcus erythropolis* W2 [106]. AiiD irreversibly breaks the amide bond between the lactone portion of the AHL molecule and the acyl chain, releasing the free pentameric ring lactone and corresponding fatty acid [103,105]. AHL oxidoreductase was initially found in a *Rhodococcus* sp. isolate, which converts AHLs into their 3-hydroxy-derivatives [106]. In freshwater aquaculture, co-injection of an AiiA type enzyme from *Bacillus* sp. B546 with a fish pathogen *Aeromonas hydrophila* to fishes could significantly reduce their mortality [114]. Addition of N-acyl homoserine lactone-degrading bacterial enrichment cultures (ECs) to the giant freshwater prawn *Macrobrachium rosenbergii* larviculture could protect larval prawns from *Vibrio harveyi* infection, which increased the survival and quality of larval prawns with no side effect on larval growth [115].

To date, only one type of enzyme was found capable of degrading DSF signals. RpfB, a homologue of the fatty acyl coenzyme A (acyl-CoA) ligase FadD in *E. coli*, was found associated with DSF turnover to generate fatty acyl-CoA in plant pathogens *Xcc* [116,117]. By using a high throughput screening approach, strong DSF degradation activity was found in a soil bacterial isolate *Pseudomonas* sp. HS-18. At least four DSF-inducible *fadD* homologues (designated as *digABCD*) were found in the genome of strain HS-18, which explains its superior DSF degradation activity [107]. Expression of the *dig* genes in *Xcc* markedly reduced the accumulation of endogenous DSF, decreased production of virulence factors, and attenuated bacterial virulence on host plants. Similarly, application of strain HS-18 as a biocontrol agent could substantially reduce the disease severity caused by *Xcc* [107].

QS inhibitors (QSIs) act through at least two mechanisms: inhibition of QS signal biosynthesis and interference with the binding of receptor proteins to QS signals. QSIs include natural compounds and synthetic analogs of QS signals. The halogenated furanone isolated from macroalga *Delisea pulchra* could be the first identified natural QSI. The furanone was found to inhibit the phenotypes under the control of the AHL QS system by accelerating the turnover of LuxR, which is the AHL signal receptor [118,119,120]. The flavonoids from *Citrus sinensis* represent another type of natural QSI, which could significantly reduce the concentration of QS signals secreted by *Yersinia enterocolitica*, and disperse the bacterial biofilm without affecting the growth of bacteria [121], but the mechanism remains unclear. Although QS antagonistic activity has been found in bacterial, fungal, plant and animal metabolites, only a small fraction of the active molecules have been isolated and identified. Another widely used approach is to synthesize QSIs based on the chemical structures of QS signals as antagonist against QS signal production or perception in pathogenic bacteria. For example, the synthetic furanone derivative compound C-30 inhibited QS-dependent biofilm formation, which increased antibiotic susceptibility of *P. aeruginosa* to tobramycin [122]. In a mouse pulmonary infection model, C-30 reduced virulence by promoting the clearance of *P. aeruginosa* through the mouse immune system [122]. 3-oxo-C_12_-(2-aminocyclohexanone), a chemically synthesized structural analog of N-oxo-C12-HSL, which is the QS signal of the *las* QS system, was able to inhibit the expression of both the AHL synthase genes *lasI* and *rhlI*, pyocyanin and elastase production, and biofilm formation [123]. In addition, some chemicals structurally unrelated to the QS signals may have more efficient QSI activity, with more stable structure and lower toxicity. For example, a triphenyl substance called TP-5 could effectively inhibit the AHL-regulated phenotypes of *P. aeruginosa* [124], but its mechanism of action needs to be further elucidated. Treatment of *P. aeruginosa* with the QSI compound 4-nitro-pyridine-N-oxide (4-NPO) was found to decrease the expression of over 30% of the QS-regulated genes, and significantly reduced the mortality of its infection model *Caenorhabditis elegans* [125].

QS signals are small hapten molecules which, when combined with a larger carrier such as a protein, might elicit the production of specific antibodies capable of binding to the signal itself. Based on this principle, a specific monoclonal antibody against AHLs were generated [126], which could reduce virulence gene expression and virulence factor production by neutralizing AHL signaling activity in *P. aeruginosa*.

Similar to QQ approaches, chemical inhibitors and antibodies were found capable of blocking pathogen–host cell–cell communications. Considering the importance of Spd and Spm signals in the regulation of the T3SS and AMR in *P. aeruginosa* [62,94], a range of Spd and Spm derivatives were synthesized on the hypothesis that certain derivatives could block the influx of the polyamine signals to bacterial cells through the SpuDEFGH transporter. One of the derivatives, R101-Spm was found capable of effectively inhibiting T3SS gene expression and significantly reducing the bacterial virulence [127]. By linking Spm to a protein carrier as antigen, Wang et al. designed and prepared a specific high-affinity monoclonal antibody against the Spd and Spm signals produced by host cells, and animal test showed that this antimicrobial agent was highly effective in control of *P. aeruginosa* infection in mice [108].

## 8. Conclusions and Future Prospective

Evidence has been accumulating that microbial QS systems and host–pathogen communication systems could play key roles in the regulation of not only virulence but also AMR. Therefore, targeting microbial cell–cell communication systems might serve two important purposes: infection control and AMR alleviation. Currently, only a few model microorganisms have been relatively well characterized. The chemical communication systems and their signaling mechanisms in many pathogens are yet to be further investigated. In particular, we are quite ignorant of how AMR mechanisms are modulated by chemical communication systems. For effective control of microbial infections and for curbing the rapid emergence of AMR problems, discovery of microbial cell–cell chemical communication systems appears to provide a brand new prospective. To fully utilize and exploit its potentials, future research could focus on the following areas.

Continued effort is warranted to discover and identify new QS and pathogen–host cell communication systems. To date, only a few families of conserved cell–cell communication signals have been identified, including AHL, DSF and polyamine signals. Given the genetic similarity of many microorganisms, it is expected that more such conserved signals could be used by different bacterial species. In addition, it is also highly likely that microbes may utilize certain unique signals to regulate a set of specific genes. Identification of new signals would be essential for understanding their biological functions and regulatory mechanisms.

Detailed analysis and understanding of microbial cell–cell communication network is required. It should be noted that microbial QS systems and their regulatory networks are complex and complicated. Evidence shows that microorganisms commonly contain more than one QS system in the regulation of bacterial virulence as well as AMR. For example, the virulence of *Salmonella enterica*, a primary enteric pathogen, is controlled by two QS systems [128]. The quorum-sensing network in *P. aeruginosa* includes multiple signaling systems, including Las, Rhl, PQS, and IQS, which form a hierarchical network that collectively regulates a variety of cellular phenotypes such as virulence factors and drug resistance [129,130]. In addition, the pathogen also responds to polyamine signals to activate the expression of T3SS genes (58). We are still very ignorant of how different chemical communication systems interact with each other and whether and how other environmental cues could influence their biological functions. Without understanding the all-round information of the chemical communication systems contributing to virulence and AMR regulation, it could be hard to develop effective and satisfactory QQ strategies against QS and host–pathogen signaling-mediated infections and AMR.

Comprehensive studies are needed on the mechanism of action, side effect and biosafety of QQ enzymes and QSIs. QQ could not only block the QS-regulated infection, but might also contribute to reducing bacterial drug resistance, which is a new strategy for anti-infection and anti-AMR therapy. However, the specificity, biosafety and mechanism of action of quorum quenchers should be fully investigated. In addition, the potentials and usage methods of QQ agents in preventive and therapeutic applications need to be investigated and evaluated. Furthermore, determination of the impact of QQ on natural microbial communities remains a key point for future research. Understanding whether QQ in nature is used for competition or cooperation and how it affects microbial interactions in host and non-host environments may provide useful insights for the prevention and control of human microbial diseases and alleviation of AMR problems.

## Figures and Tables

**Figure 1 antibiotics-09-00779-f001:**
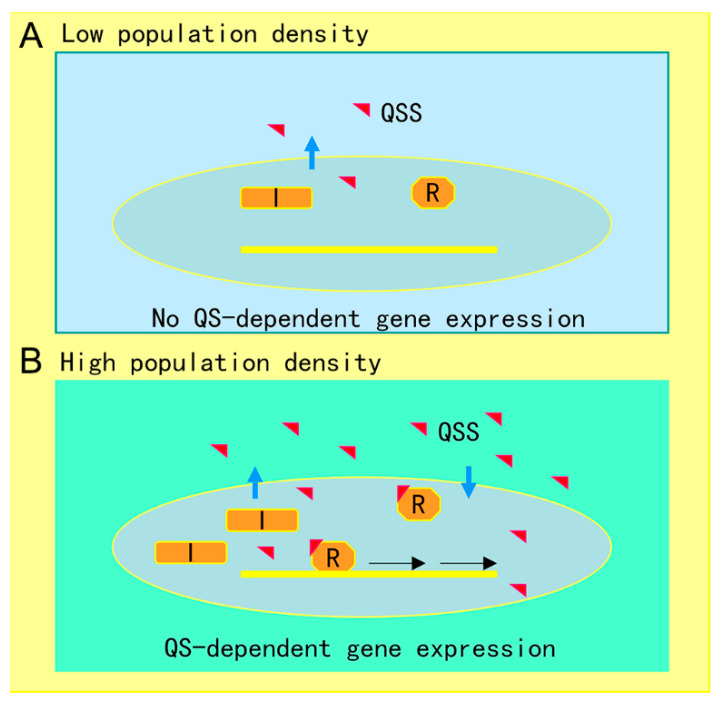
Illustration of quorum sensing (QS) mechanism in microorganisms. (**A**) At low cell population density, bacterial cells produce limited QS signal (QSS) which cannot trigger QS-dependent gene expression. (**B**) Along with bacterial growth, accumulated QSS interacts with and hence activates its cognate receptor to induce virulence factor production, biofilm formation, and generation of efflux pumps, which aid the pathogen survival in the host by counteracting various possible stresses, including immune responses and antibiotics. Symbols: I represents the QS signal synthase, R is the receptor, and the red triangle indicates QS signal.

**Figure 2 antibiotics-09-00779-f002:**
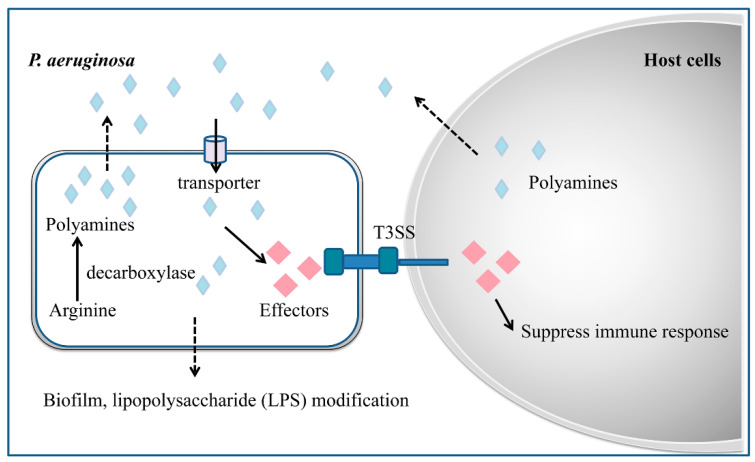
The role of the polyamine chemical communication system in modulation of virulence and antibiotic resistance in *P. aeruginosa*. Microorganisms synthesize polyamines by themselves using arginine as the substrate or can uptake polyamines from host cells or the surrounding environment through specific transporters, which induce expression of the genes encoding virulence factor production, biofilm formation, and coordinate responses to oxidative stresses.

**Table 1 antibiotics-09-00779-t001:** Strategies for blocking microbial cell-cell communication systems.

Type of Quorum Quenching Strategies	Example	Action Mechanism	Target	Reference
Signal degradation or modification	AHL-lactonases (AiiA)	Hydrolyzes the lactone ring of AHL molecules	Acyl homoserine lactone (AHL) family quorum sensing(QS) signals	[101]
	AHL-acylases (AiiD)	Breaks the amide bond between the lactone portion of the AHL molecule and the acyl chain		[105]
	AHL oxidoreductases	Targets the acyl side chain by oxidative but not degradation		[106]
	Dig 1–4	Similar to their homologue FadD, which catalyzes the esterification of long-chain fatty acids into metabolically active coenzyme A thioesters	Diffusible signaling factor (DSF) family QS signals	[107]
Signal neutralization	Antibody Mab 4E4	Inactivation of Spd and Spm by specific binding	Polyamine signals	[108]
Signal synthase inhibitor	J8-C8	Inhibition of QS signal biosynthesis	QS signal synthase	[109]
Signal receptor inhibitor	E9C-3oxoC6	Interference with the binding of receptor proteins to QS signals	QS signal receptor	[109]

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
