# Peer review of "The Roles of Microbial Cell-Cell Chemical Communication Systems in the Modulation of Antimicrobial Resistance"

_antibiotics, 2020, doi:10.3390/antibiotics9110779_

Round 1

Reviewer 1 Report

The review by Huang et al. presents a limited overview on systems mediating AMR in bacteria, with an emphasis on quorum sensing as a potential therapeutic target for infection control. Although the Author’s structured the piece in an adequate manner, it is evident that this review lacks the relevant in depth discussion of AMR systems, particularly regarding how QS can be successfully exploited as a therapeutic to control the spread of AMR. My comments are outlined as follows:

Comments

Line 32 – This sentence is not accurate. The term ‘basically helpless’ discounts the effects of the host innate and adaptive immune response.

Line 34 – This is still the case.

Line 35 – ‘Had’ not ‘has.

Line 46 – The term ‘serious’ is ambiguous when used in this context.

Line 49 – What is the current burden of AMR, both financially and in adjusted life years?

Line 51 – ‘Lack of antibiotic stewardship’ instead of ‘abuse’.

Line 61 – Should be written as ‘which efflux multiple antibiotics…’

Line 96 – Not all antibiotics target metabolic processes. i.e. polymyxin and lipopeptides class antibiotics.

Line 98 – ‘metabolic’ is not the correct term to be used here.

Line 105 – ‘Removal’ not ‘passivation’.

Line 123 – This paragraph appears to be border-line plagiarism extracted from the following paper without citation - Ding et al 2020 doi: 10.3390/microorganisms8030425

Huang et al - In addition, bacterial biofilm constitutes a systematic and complex AMR mechanism, which has at least the three main principles: (1) biofilm itself is an effective drug barrier, which can significantly reduce the permeability and sensitivity of antibiotics [39,40]; (2) the special microenvironment within the biofilm, causing heterogeneity of the bacteria cells within biofilm matrix, which facilitates antibiotic tolerance of bacteria [41]; (3) the extreme environment within the biofilm accelerates the production of persisters and promotes AMR development in bacteria [42].

Ding et al - The drug resistance produced by bacterial biofilm is a systematic and complex drug resistance mechanism. The principle of drug resistance has at least the following three points [34,35,36]: (1) The biofilm itself is an effective drug barrier, which can significantly reduce it antibiotic permeability. Bacteria are connected to each other through proteins and DNA, especially extracellular polysaccharides, forming an insurmountable barrier, which can greatly reduce the permeability of antibiotic drugs and improve the survival rate of bacteria in the biofilm. (2) The special microenvironment in the biofilm makes the bacteria in the membrane produce heterogeneity and regulates the antibiotic resistance of the bacteria. The study found that the concentrations of nutrients and bacterial secretions in different areas of the biofilm were not the same, which led to the inconsistent growth status of the bacterial bodies in different areas of the biofilm, that is, the heterogeneity of the bacteria, which led to different levels of drug resistance bacterial cells. (3) The extreme environment outside the biofilm promotes drug resistance within the membrane.

Line 221 – Terms have been abbreviated, then used in full throughout the manuscript. i.e. spermidine and spermine.

Figures – All figures (figures 1-4) are of extremely poor quality. They are inadequately illustrated and do not aid the reader in comprehending each AMR or QS process. E.g. In figure 1, ‘decreased permeability’ is exemplified by an orange box! Also, each respective figure legend is also lacking an in-depth description. Figures do not complement the review, rather they are included for the sake of it.

General comment: It is not clear to the reader how QS can be successfully exploited therapeutically to overcome the burden of AMR. Much of the focus has been applied to a random set of pathogens, and not those which significantly contribute to community- and hospital-acquired infection (with exception to E. coli and P. aeruginosa). Line 414 is one of the few instances where therapeutic potential is outlined.

Reviewer 2 Report

This review focused on the modulation of antimicrobial resistance (AMR) by bacterial cell-to-cell signaling and the potential of quenching the microbial signaling as a novel strategy for AMR pathogen treatment. With the overuse of antibiotics and the mutation of microorganisms, microorganisms have developed a wide variety of resistance mechanisms. AMR phenomenon has become a major threat to human life and health. Humans urgently need new weapons to fight microbial drug resistance. In recent years, an increasing number of studies have identified an important role for microbial cell-to-cell chemical communication systems in the regulation of different AMR mechanisms. These systems include quorum sensing (QS) and pathogen-host communication mechanisms, which are signaling molecules produced by microbes and host cells, were identified to regulate drug efflux pump, biofilm matrix formation and other AMR associated pathways. This article also summarizes some recent progress on the strategy for quenching the bacterial chemical communication systems to combat the problem of microbial drug resistance. Although bacterial chemical communication system has been well studied in several model strains, we need to discovery new communication systems, clarify the regulatory network between multiple signals and comprehensively assess the biosafety of quorum quenching agents. Overall, the manuscript is well-written and should be of potential interest in the field. The reviewer would recommend its publication upon some minor revisions.

Minor comments:

1` L59-60. two-component systems (TCSs)

2` L62-63. two-component systems (TCSs)

3` L100. “and” seems unnecessary

4` L116-120. The opening sentence is simply too long and difficult to follow.

5` L144. The genus of “Photobacterium fischeri” had been reclassified to Aliivibrio

6` L206-207. “the protein kinase RpfC self-phosphorylates in response to DSF signals” should be revised to “In response to DSF signals leads to the activation of the protein kinase RpfC by self-phosphorylation, which…”

7` L395. “agonists” should be revised to “antagonist”

8` L441. “QS systems” seems unnecessary

Reviewer 3 Report

This a good and timely review on the relation between microbial cell-cell communication and antibiotic resistance.

Please find the comments in the attachment.

Round 2

Reviewer 1 Report

The authors now provide a significantly improved manuscript. A clearer rationale is now described regarding how QS can be potentially utilised as a therapeutic to control the spread of AMR. I only have a few minor comments which require addressing prior to acceptance:

Minor comments:

Line 52: should read "with up to 100 trillion US dollars in global economic losses". 

Line 96: should read "through either inhibiting microbial primary metabolic processes or compromising pathogen cell-membrane integrity".

Line 98: There is a full-stop after the word "processes" which should be removed.

Line 102: "can", not "could".

Line 116: "biofilm producing bacteria can...". "Biofilm bacteria" is not correct. 

119: "can alter" not "altered". 

Line 180, Figure 1: If the figure is to be separated into two panels, label them as A and B. 
